# New relevance and significance measures to replace p-values

**Werner A. Stahel** *

Seminar for Statistics, ETH, Zurich, Switzerland

* stahel@stat.math.ethz.ch

## Abstract

The p-value has been debated exorbitantly in the last decades, experiencing fierce critique, but also finding some advocates. The fundamental issue with its misleading interpretation stems from its common use for testing the unrealistic null hypothesis of an effect that is precisely zero. A meaningful question asks instead whether the effect is *relevant*. It is then unavoidable that a threshold for relevance is chosen. Considerations that can lead to agreeable conventions for this choice are presented for several commonly used statistical situations. Based on the threshold, a simple quantitative measure of relevance emerges naturally. Statistical inference for the effect should be based on the confidence interval for the relevance measure. A classification of results that goes beyond a simple distinction like "significant / non-significant" is proposed. On the other hand, if desired, a single number called the "secured relevance" may summarize the result, like the p-value does it, but with a scientifically meaningful interpretation.

**Data Availability Statement:** All relevant data are within the manuscript.

**Funding:** The author(s) received no specific funding for this work.

## 1 Introduction

The p-value is arguably the most used and most controversial concept of applied statistics. Blume *et al.* [1] summarize the shoreless debate about its flaws as follows: "Recurring themes include the difference between statistical and scientific significance, the routine misinterpretation of non-significant p-values, the unrealistic nature of a point null hypothesis, and the challenges with multiple comparisons." They nicely collect 14 citations, and I refrain from repeating their introduction here, but complement the analysis of the problem and propose a solution that both simplifies and extends their's.

The basic cause of the notorious lack of reliability of empirical research, notably in parts of social and medical science, can be found in the failure to ask scientific questions in a sufficiently explicit form, and the p-value problem is intrinsically tied to this flaw. Here is my argument.

Most empirical studies focus on the effect of some treatment, expressed as the difference of a target variable between groups, or on the relationship between two or more variables, often expressed with a regression model. Inferential statistics needs a probabilistic model that describes the scientific question. Usually, this is a parametric model in which the effect of

**Competing interests:** The authors have declared that no competing interests exist.

interest appears as a parameter. The question is then typically specified as: "Can we prove that the effect is not zero?"

## 1.1 The Zero Hypothesis Testing Paradox

This is, however, not a scientifically meaningful question. When a study is undertaken to find some difference between groups or some influence between variables, the *true* effect—e.g., the difference between two within group expected values—will never be precisely zero. Therefore, the strawman null hypothesis of zero true effect (the "zero hypothesis") could in almost all reasonable applications be rejected if one had the patience and resources to obtain enough observations. Consequently, the question that is answered mutates to: "Did we produce sufficiently many observations to prove the (alternative) hypothesis that was true on an apriori basis?" This does not seem to be a fascinating task. I call this argument the "Zero Hypothesis Testing Paradox." This paradox has been stated prominently as a problem in the philosophy of science over fifty years ago in a highly cited long paper by Meehl [2].

The problem with the p-value is thus that it is the output of testing an unrealistic null hypothesis and thereby answers a nonsensical scientific question. (Note that the proposal to lower the testing level from 5% to 0.5% by Benjamin *et al.* [3] is of no help in this respect, see also [4]).

A sound question about an effect is whether it is large enough to be *relevant*. In other words: Without the specification of a threshold of relevance, the scientific question is void.

Scientists have gladly avoided the determination of such a threshold, because they felt that it would be arbitrary, and have jumped on the train of "Null Hypothesis Significance Testing" that was offered cheaply by statistics. Let us be clear: Avoiding the choice of a relevance threshold means avoiding a scientifically meaningful question.

Given the relevance threshold, the well-known procedures can be applied not only for testing the null hypothesis that the effect is larger than the threshold against the alternative that it is smaller, but also vice versa, proving statistically that the effect is negligible. The result can of course also be ambiguous, meaning that the estimate is neither significantly larger nor smaller than the threshold. I introduce a finer distinction of cases in Section 2.3.

These ideas are well-known under the heading of equivalence testing, and similar approaches have been advocated in connection with the p-value problem, like the "Two One-Sided Tests (TOST)" of Lakens [5], the "Second Generation p-value (SGPV)" by Blume *et al.* [1], or the "Minimum Effect Size plus p-value (MESP)" by Goodman *et al.* [6]. The threshold has been labelled "Smallest Effect Size Of Interest (SESOI)", "Minimum Practically Significant Distance (MPSD)," or the limit of the "Region of Practical Equivalence (ROPE)." [7] I come back to these concepts in Section 2.2. Kruschke [7] proposes a Bayesian approach that is very similar to the ideas presented in this paper, see Section 2.3.

Using confidence intervals instead of p-values or even "yes-no" results of null hypothesis tests provides the preferable, well-known alternative to null hypothesis testing for drawing adequate inference. Each reader can then judge a result by checking if his or her own threshold of relevance is contained in the interval. Providing confidence intervals routinely would have gone a long way to solving the problem. I come back to this issue in the Discussion (Section 7).

Most probably, the preference to present p-values rather than confidence intervals is due to the latter's slightly more complicated nature. In their usual form, they are given by two numbers that are not directly comparable between applications. I will define a single number, which I call "significance," that characterizes the essence of the confidence interval in a simple and informative way.

In "ancient" times, before the computer produced p-values readily, statisticians examined the test statistics and then compared them to tables of "critical values." In the widespread case that the t-test was concerned, they used the t statistic as an informal quantitative measure of significance of an effect by comparing it to the number 2, which is approximately the critical value for moderate to large numbers of degrees of freedom. This will also shine up in the proposed significance measure.

In a similar way, the proposed quantitative measure of relevance divides the effect by a meaningful threshold, and a value above 1 indicates a relevant effect. In contrast to significance, relevance is a parameter of the model. As such, it is estimated on the basis of the observations, and a confidence interval for it should be determined. The lower end of this confidence interval will be proposed as a single most interpretable characteristic.

This quantitative measure of relevance is most generally interpretable if applied to a suitable way of expressing the effect of interest. This leads to standardizing or transforming model parameters in order to determine an appropriate "effect scale." The idea of effect scale is partly parallel and partly alternative to the "effect size" definitions that are popular in quantitative psychology [8, 9]. Section 3 proposes these scales for the most commonly used stattistical models.

The suitable relevance threshold should be determined in the context of the scientific question. As a professional statistician, I prefer to leave the choice to the scientist who formulates this question. As a consultant, I appreciate the hurdle that this desideratum poses to the practical application of the concept of the relevance measure and give in to providing a recommendation that can be used as a starting point and default (Section 5).

## 2 Definitions

The simplest case for statistical inference is the estimation of a constant based on a sample of normal observations. It directly applies to the estimation of a difference between two treatments using paired observations. I introduce the new concepts first for this situation. The application of the concepts for typical situations—comparison of two samples, estimation of proportions, simple regression and correlation—will be discussed in Section 3 and extended to a general parametric model and to multiple regression in Section 4.

### 2.1 The generic case

Consider a sample of $n$ statistically independent observations $Y_i$ with a normal distribution,

$$Y_i \sim \mathcal{N}(\mu, \sigma^2) .$$

The interest is in the effect parameter $\vartheta = \mu$, and more specifically, knowing whether $\vartheta$ is different from 0 in a relevant manner, where relevance is determined by the relevance threshold $\zeta > 0$. Thus, one wants to summarize the evidence for the hypotheses

$$H_1 : \ \vartheta > \zeta \quad \text{against} \quad H_0 : \ \vartheta \le \zeta .$$

(The symbol $\zeta$, pronounced "zeta," delimits the "zero" hypothesis).

**2.1.1 One sided.** I consider a one-sided hypothesis here. In practice, only one direction of the effect is usually plausible and/or of interest. Even if this is not the case, the conclusion drawn will be one-sided: If the estimate turns out to be significant according to the two-sided test for 0 effect, then nobody will conclude that "the effect is different from zero, but we do not know whether it is positive or negative." Therefore, in reality, two one-sided tests are conducted, and technically speaking, a Bonferroni correction is applied by using the level $\alpha/2 = 0.025$ for each of them. Thus, I treat the one-sided hypothesis and use this testing level.

The point estimate and confidence interval are

$$\hat{\vartheta} = \bar{Y} = \frac{1}{n}\sum_i Y_i \, , \qquad \mathrm{CI}_\vartheta = \hat{\vartheta} \pm \hat{\omega} \, , \qquad \hat{\omega} = q\sqrt{\hat{V}/n} \, ,$$

where $\hat{V}$ is the empirical variance of the sample, $\hat{V} = \frac{1}{n-1}\sum_i (Y_i - \bar{Y})^2$, and $q$ is the $1-\alpha/2 = 0.975$ quantile of the appropriate $t$ distribution. Thus, $\hat{\omega}$ is half the width of the confidence interval and equals the standard error, multiplied by the quantile.

**2.1.2 Remark.** The choice of the test level, $\alpha$, is arbitrary in principle. In some fields, $\alpha = 0.01$ is common, but $\alpha = 0.05$ is clearly the most popular choice and ubiquitous in many fields. It is straightforward to adjust all concepts introduced here to any $\alpha$.

**2.1.3 Significance.** The proposed significance measure compares the difference between the estimated effect and the relevance threshold with the half width of the confidence interval,

$$\mathrm{Sig}_\zeta = (\hat{\vartheta} - \zeta)/\hat{\omega} \, . \tag{1}$$

The effect is statistically significantly larger than the threshold if and only if $\mathrm{Sig}_\zeta > 1$.

Significance can also be calculated for the common test for zero effect, $\mathrm{Sig}_0 = \hat{\vartheta}/\hat{\omega}$. This quantity can be listed in computer output in the same manner as the p-value is given in today's programs, without a requirement to specify $\zeta$. It is much easier to interpret than the p-value, since it is, for a given precision expressed by $\hat{\omega}$, proportional to the estimated effect $\hat{\vartheta}$. Furthermore, a standardized version of the confidence interval for the effect is $\mathrm{Sig}_0 \pm 1$,

$$\mathrm{Sig}_0 \pm 1 = \mathrm{CI}_\vartheta/\hat{\omega} \, , \qquad \mathrm{CI}_\vartheta = \hat{\vartheta}(1 \pm 1/\mathrm{Sig}_0) \, . \tag{2}$$

Nevertheless, it should be clear from the Introduction that $\mathrm{Sig}_0$ should only be used with extreme caution, since it does not reflect relevance.

**2.1.4 Relevance.** An extremely simple and intuitive quantitative measure of relevance is the effect, expressed in $\zeta$ units,

$$\mathrm{Rl} = \vartheta/\zeta \, . \tag{3}$$

Its point and interval estimates are

$$\mathrm{Rle} = \hat{\vartheta}/\zeta \qquad \text{and} \qquad [\mathrm{Rls}, \mathrm{Rlp}] \, , \quad \text{where}$$
$$\mathrm{Rls} = \mathrm{Rle} - \hat{\omega}^* \, , \quad \mathrm{Rlp} = \mathrm{Rle} + \hat{\omega}^* \, , \qquad \hat{\omega}^* = \hat{\omega}/\zeta \, . \tag{4}$$

The lower end of the confidence interval is called the "secured relevance," Rls, and the upper end, the "potential relevance," Rlp. The effect is called *relevant* if $\mathrm{Rls} > 1$, that is, if the estimated effect is significantly larger than the threshold.

The estimated relevance Rle is related to $\mathrm{Sig}_\zeta$ by

$$\mathrm{Sig}_\zeta = (\mathrm{Rle} - 1)/\hat{\omega}^* \, , \qquad \mathrm{Rle} = \mathrm{Sig}_\zeta \, \hat{\omega}^* + 1 \, .$$

Fig 2 shows several cases of relations between the confidence interval and the effects 0 and $\zeta$, which can be translated into categories that help interpret results, see Section 2.3.

**2.1.5 Example: Student's sleep data.** Student [10] illustrated his t-test with data measuring the extra sleep evoked by a sleep enhancing drug in 10 patients. The numbers in minutes are −6, 6, 48, 66, 96, 114, 204, 264, 276, 330. Their mean is $\hat{\vartheta} = \bar{Y} = 140$. The p-value for testing the hypothesis of no prolongation is 0.5% and the confidence interval extends from 54 to 226. The zero significance is obtained from $V = 14$,–432, $n = 10$ and $q = 2.26$ with $\hat{\omega} = 2.26\sqrt{14{,}432/10} = 86$ as $\mathrm{Sig}_0 = 140/86 = 1.63$.

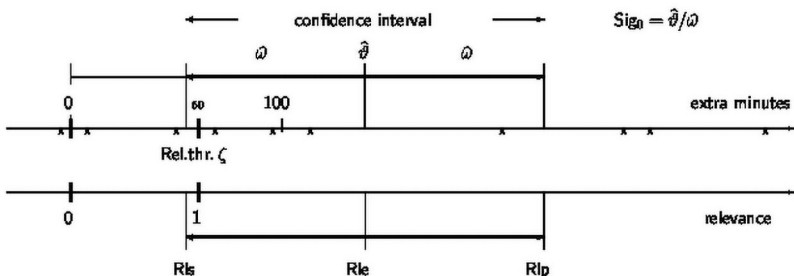

**Fig 1. Estimate, confidence interval and relevance for the sleep data.**

If the relevance threshold is one hour of extra sleep, $\zeta = 60$, then $\text{Sig}_\zeta = 80/86 = 0.93$, and the gain is not significantly relevant. This is also seen when calculating the relevance and its confidence interval, Rle $= 140/60 = 2.33$ and Rls $= 2.33 - 86/60 = 54/60 = 0.90$, Rlp $= 2.33 + 86/60 = 226/60 = 3.76$. It remains therefore unclear whether the sleep prolongation is relevant. Fig 1 shows the results graphically.

## 2.2 Related concepts

**2.2.1 Two One-Sided Tests (TOST).**  Lakens [5] focusses on testing for a negligible effect, advocating the paradigm of equivalence testing. He considers an interval of values that are negligibly different from the point null hypothesis, also called a "thick" or "interval null" [1, 6]. If this interval is denoted as $|\vartheta| \leq \zeta$, there is a significantly negligible effect if both hypotheses $\vartheta > \zeta$ and $\vartheta < -\zeta$ are rejected using a one-sided test for each of them. A respective p-value is the larger of the p-values for the two tests.

I have argued for a one-sided view of the scientific problem. With this perspective, the idea reduces to the *one* one-sided test for a negligible effect with significance measure $-\text{Sig}_\zeta$.

**2.2.2 Second Generation P-value.**  The "Second Generation P-Value" SGPV $P_\delta$ has been introduced by Blume *et al.* [1, 11]. In the present notation, $\zeta$ is their $\delta$. The definition of $P_\zeta$ starts from considering the length $O$ of the overlap of the confidence interval with the interval defined by the composite null hypothesis $H_0$. Assume first that $\hat{\vartheta} > 0$. Then, the overlap measures $O = 2\hat{\omega}$ if the confidence interval contains the "null interval," that is, if $\hat{\vartheta} + \hat{\omega} < \zeta$, and otherwise, $O = \zeta - (\hat{\vartheta} - \hat{\omega})$, or 0 if this is negative.

The definition of $P_\zeta$ distinguishes two cases based on comparing $\hat{\omega}$ to the threshold $\zeta$. If $\hat{\omega} < 2\zeta$, $P_\zeta = 0$ if there is no overlap, and $P_\zeta = 1$ for complete overlap, $O = 2\hat{\omega}$. In between, the SGPV is the overlap, compared to the length of the confidence interval,

$$P_\zeta = \frac{O}{2\hat{\omega}} = \frac{\zeta - (\hat{\vartheta} - \hat{\omega})}{2\hat{\omega}} = \frac{\zeta - \hat{\vartheta}}{2\hat{\omega}} + \frac{1}{2} = \frac{1}{2}\left(1 - \text{Sig}_\zeta\right) .$$

In this case, then, $P_\zeta$ is a rescaled, mirrored, and truncated version of the significance at $\zeta$.

Here, I have neglected a complication that arises when the confidence interval covers values below $-\zeta$. The definition of $P_\zeta$ starts from a two-sided formulaton of the problem, $H_0$: $|\vartheta| < \zeta$. Then, the confidence interval can also cover values below $-\zeta$. In this case, the overlap decreases and $P_\zeta$ changes accordingly.

The definition of $P_\zeta$ changes if the confidence interval is too large, specifically, if its length exceeds $2\zeta$. This comes again from the fact that it was introduced with the two-sided problem in mind. In order to avoid small values of $P_\zeta$ caused by a large denominator $2\hat{\omega}$ in this case, the length of the overlap $O$ is divided by twice the length $2\zeta$ of the "null interval," instead of the

length of the confidence interval, $2\hat{\omega}$, $P_\zeta = O/(4\zeta)$. Then, $P_\zeta$ has a maximum value of 1/2, which is a deliberate consequence of the definition, as this value does not suggest a "proof" of $H_0$. For a comparison of the SGPV with TOST, see [12].

If the overlap is empty, $P_\zeta = 0$. In this case, the concept of SGPV is supplemented with the notion of the "$\delta$ gap,"

$$\text{Gap}_\zeta = (\hat{\vartheta} - \zeta)/\zeta = \text{Rle} - 1 \ .$$

Since the significance and relevance measures are closely related to the Second Generation P-Value and the $\delta$ gap, one might ask why still new measures should be introduced. Here is why:

- An explicit motivation for the SGPV was that it should resemble the traditional p-value by being restricted to the 0-1 interval. I find this quite undesirable, as it perpetuates the misinterpretation of $P$ as a probability. Even worse, the new concept is further removed from such an interpretation than the old one, for which the problem "Find a correct statement including the terms p-value and probability" still has a (rather abstract) solution.

- The new p-value was constructed to share with the classical one the property that small values signal a large effect. This is a counter-intuitive aspect that leads to confusion for all beginners in statistics. In contrast, larger effects lead to larger significance (and, of course, larger relevance).

- Taking these arguments together, the problems with the p-value are severe enough to prefer a new concept with a new name and more direct and intuitive interpretation over advocating a new version of p-value that will be confused with the traditional one.

- The definition of the SGPV is unnecessarily complicated, since it is intended to correspond to the two-sided testing problem, and only quantifies the undesirable case of ambiguous results. It deliberately avoids to quantify the strength of evidence in the two cases in which either $H_0$ or $H_1$ is accepted.

## 2.3 Classification of results

There is a wide consensus that statistical inference should *not* be reported simply as "significant" or "non-significant." Nevertheless, communication needs words. I therefore propose to distiguish the cases that the effect is shown to be relevant (Rlv), that is, $H_1: \vartheta > \zeta$ is "statistically proven," or negligible (Ngl), that is, $H_0: \vartheta \leq \zeta$ is proven, or the result is ambiguous (Amb), based on the significance measure $\text{Sig}_\zeta$ or on the secured and potential relevance Rls and Rlp (Rls $> 1$ for Rlv, Rlp $< 1$ for Ngl and Rls $\leq 1 \leq$ Rlp for Amb).

**2.3.1 Remark.** Kruschke [7] distinguishes the same cases on the basis of the Bayesian Posterior Highest Density Interval and calls them "Reject Null Value," "Accept Null Value," and "Undecided"—examining, however, a two-sided "Region of Practical Equivalence."

For a finer classification, the significance for a zero effect, $\text{Sig}_0$, is also taken into account. This may even lead to a contradiction (Ctr) if the estimated effect is significantly negative. Fig 2 shows the different cases with corresponding typical confidence intervals, and Table 1 lists the respective significance and relevance ranges. Similar figures have appeared in [1, Fig 2] and [6, Fig 1] and before, with different interpretations.

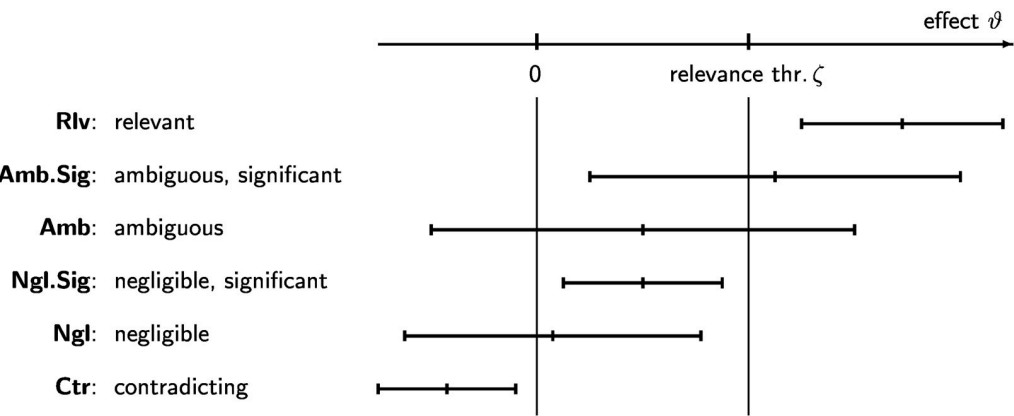

**Fig 2. Classification of cases based on a confidence interval and a relevance threshold.**

## 3 Generalization to more models

### 3.1 The two-sample problem

The usual model for comparing two treatments arises when $x_i = 1$ if observation $i$ received one treatment, and $x_i = -1$ for the other treatment. (The code for the second group is $-1$ instead of 0 since this choice fits better with the standardized coefficient of linear regression to be treated below.) Then,

$$Y_i \sim \mathcal{N}(\mu_0 + \theta x_i, \sigma^2) .$$

The effect parameter $\theta$ is the half the difference of expected values between the two groups, whereas $\mu_0$ and $\sigma$ are nuisance parameters.

**3.1.1 Effect scale.** In several models, it appears useful to consider a transformed version of the parameter of interest as the effect, since the transformation leads to a more generally interpretable measure and may have more appealing properties, as in the next subsection. Therefore, the original parameter of interest is denoted as $\theta$ or as popular in the model, and the transformed version will be considered as the effect, $\vartheta = g(\theta)$.

**3.1.2 Standardization.** In the case of two samples, it is very popular to standardize the difference between the groups in order to make it independent of any unit of measurement, leading to Cohen [13]'s $d$, which is, in the present notation, $d = 2\theta/\sigma$. In the same way, the effect size is introduced here as

$$\vartheta = \theta/\sigma = d/2 . \tag{5}$$

**Table 1. Classification of cases defined by ranges of significance and relevance measures.** $s$ and $r$ are the place holders for the column headings.

| Case | $\text{Sig}_0$ | $\text{Sig}_\zeta$ | Rls | Rlp |
|---|---|---|---|---|
| Rlv | $s \gg 1$ | $s > 1$ | $r > 1$ | $r \gg 1$ |
| Amb.Sig | $s > 1$ | $-1 < s < 1$ | $0 < r < 1$ | $r > 1$ |
| Amb | $-1 < s < 1$ | $-1 < s < 1$ | $r < 0$ | $r > 1$ |
| Ngl.Sig | $s > 1$ | $s < -1$ | $0 < r < 1$ | $0 < r < 1$ |
| Ngl | $-1 < s < 1$ | $s < -1$ | $r < 0$ | $0 < r < 1$ |
| Ctr | $s < -1$ | $s \ll -1$ | $r \ll 0$ | $r < 0$ |

Note that standardization with the variation $\sigma$ of the target variable within groups makes good sense if $\sigma$ measures the natural variation between observation units. It is less well justified if it includes measurement error, since this would change if more precise measurements were obtained, for example, by averaging over several repeated measurements. In this case, the standardized effect is not defined by the scientific question alone, but also by the study design.

Even though $d$ and $\vartheta$ have been introduced in the two samples framework, they also apply to a single sample, since the effect in this case is the difference between its expected value and a potential population that has an expectation of zero. In this case, $\vartheta = d$. Remember that the effect is defined as a function of parameters, not of their estimates.

Coming back to the paired observation case (Section 2), note that the standard deviation measures the variability of differences rather than of the observations of the variable under study, and this will often be inappropriate. This shows that standardization may be misleading in the sense that the standardized effect does not reflect an aspect of the scientific question alone but also depends on the study design and the estimator used (see [14], p.396).

**3.1.3 Log scale.**   Most quantitative target variables in the exact and life sciences are measurements that cannot be negative, and for which differences are naturally expressed as percentages, that is, effects are best described by proportions. Such variables have been called "amounts" by the great promotor of applied statistics John W. Tukey, and he strongly recommended to express them in terms of logarithms, calling this the "first aid transformation" for such variables. On this scale, the variables usually fulfill assumptions of equal variances, of normal or at least symmetrical distributions, and of linear relationships much better than on their original scale of measurement. In other words, such variables often show a log-normal distribution on their original scale, and effects of treatments turn out to be multiplicative. Therefore, the log transformation turns them into normally distributed variables and the effects into additive ones [15].

A further advantage of the log scale is that differences become independent of any unit of measurement, and effects are directly comparable. An increase by 5% turns into an additive effect of 0.05, and generally, an increase of $p\%$, into $\log(1 + p/100)$ (which is $\approx 1 + p/100$ for small $p$). Therefore, no standardization relating to any variabilities is needed.

**3.1.4 Log-percent.**   When using percentages or ratios, it is often arbitrary which of the two numbers is taken as the reference. If one is 25% larger than the other, then the other is 20% smaller than the first. This asymmetry is a nuisance that disappears on the log scale. Therefore, let the "log-percent" scale for relative effects be defined as $100 \cdot \vartheta$, $\vartheta = \log(\mu_1) - \log(\mu_0)$, and indicate it as, e.g., $22.3\%\ell$. For small percentages, the ordinary "percent change" and the "log-percent change" are approximately equal. The new scale has the advantage of being symmetric in the two values generating the change, and therefore, the discussion whether to use the first or the second as a basis is obsolete. A change by $100\%\ell$ equals an increase of $100\%\,(e - 1) = 171\%$ ordinary percent, or a decrease by $100\%\,(1 - 1/e) = 63\%$ in reverse direction.

**3.1.5 Inference.**   Note that all these considerations regard parameters of the model and do not depend on the methods used to estimate them from observations. Effects are estimated by replacing model parameters by estimators in their defining equations, leading to point and interval estimates.

## 3.2 Proportions

When a proportion is estimated, the model is, using $\mathcal{B}$ to denote the binomial distribution,

$$Y_i \;\sim\; \mathcal{B}(1, p)\,, \qquad S = \textstyle\sum_i Y_i \sim \mathcal{B}(n, p)\,.$$

Considering variations in the probability parameter $p$, a difference of 0.05 clearly has different

relevance along the range of values of the parameter: It may be plausible to say that a change from $p$ to $p + 0.05$ for $p = 0.5$ (i.e., from 0.5 to 0.55) is barely relevant, but if $p = 0.05$ or below, the difference is large, and for $p > 0.95$, it is even impossible.

**3.2.1 Log odds.** For good and well known reasons, probabilities are often expressed as odds or log odds, also known as the "logit transformation." Let

$$\vartheta = \text{logit}(p) = \log\left(\frac{p}{1-p}\right).$$

The difference between $p = 0.5$ and $0.55$ corresponds to a difference of 0.2 on the log odds scale. The same difference on this effect scale results between $p = 0.1$ and $p = 0.12$ and, also for smaller probabilities $p$, when one is about 20% larger than the other. In fact, for low probabilities, common in the assessment of risks, log odds turn into simple logarithms, and differences of logs correspond to relative differences on the original scale. Thus, generally, equal differences of log odds appear intuitively quite comparable in relevance on the original scale, and effects on proportions should be measured on this effect scale. (Note again that the problem of estimating the effect has not been considered yet).

**3.2.2 Logit-percent.** The idea of the log-percent scale extends to the logit scale: An effect of $\vartheta = \log(p_1/(1-p_1)) - \log(p_0/(1-p_0))$ is expressed as $100 \cdot \vartheta\%\ell$. Then, the discrepancy between $p_0 = 0.5$ and $p_1 = 0.55$ equals an effect of $100 \cdot \log(0.55/(1-0.55)) - 0 = 20.1\%\ell$.

**3.2.3 Comparing two proportions.** Log-odds are again suitable for a comparison between two proportions $p_0$ and $p_1$. They lead to the log-odds ratio,

$$\vartheta = \log\left(\frac{p_1}{1-p_1} \Big/ \frac{p_0}{1-p_0}\right) = \log(p_1/(1-p_1)) - \log(p_0/(1-p_0)).$$

## 3.3 Simple regression and correlation

**3.3.1 Normal response.** In applications of the common simple regression model,

$$Y_i = \alpha + \beta x_i + \epsilon_i, \qquad \epsilon_i \sim \mathcal{N}(0, \sigma^2),$$

the slope is almost always the parameter of interest, $\theta = \beta$. It measures the change in the target variable $Y$ evoked by a change $\delta_X = 1$ in the input variable $X$.

For a standardized measure of the effect, a suitable step $\delta_X$, independent of $X$'s unit of measurement, should be chosen, and the change in $Y$ should also be standardized. The well known standardized coefficient $\beta^*$ uses the empirical standard deviation $s_X$ as $\delta_X$ and the (marginal) standard deviation $s_Y$ for the standardization, $\hat{\beta}^* = \hat{\beta} s_X/s_Y$. Here, I prefer to measure the effect in units of the error standard deviation $\sigma$, since $s_Y$ is not a model parameter (unless $X$ is modeled as a random variable), but depends on the set of $x_i$'s for which observations are obtained, that is, on the design. Therefore, the effect measure is

$$\vartheta = \beta\, \delta_X/\sigma.$$

In the case of a binary input variable $X$, the regression model is equivalent to the two groups problem treated above, and setting $\delta_X = 1$ leads to the effect measure introduced there (if the two values are coded as 1 and −1). For $X$'s with more values and in the absence of a more natural alternative, the "standard step" $\delta_X$ should be proportional to a measure of scatter of $X$ values, and $s_X$ is the straightforward choice. Note that for a binary variable with equal group sizes and codes 1 and −1, $s_X = 1$, and the two definitions match. (This was the reason for introducing these codes in the two groups case).

**3.3.2 Remark.** By setting $\delta_X = s_X$, the standardization depends on the values $x_i$ for which observations are obtained. In experiments, these are chosen by the experimenter, and the effect measure then does not describe a parameter of the process under examination alone—an undesirable feature in the spirit of this paper.

**3.3.3 Other regression models.** For a binary response variable $Y$, logistic regression provides the most well established and successful model. It reads

$$\text{logit}(P(Y_i = 1)) \quad = \quad \alpha + \beta x_i \,.$$

The parameter of interest is again $\beta$. The considerations for proportions extend directly to this model, and the effect scale is $\vartheta = \beta \delta_X$ with the same arguments for choosing $\delta_X$ as for ordinary regression. The same is true for proportional odds linear regression (POLR) for an ordered target variable.

In Poisson regression for frequency or count data, the link function connecting the linear predictor $\alpha + \beta x_i$ to the expected value of the target variable $Y$ is the logarithm, which again needs no standardization, and the same simple definition of effects scale applies. Finally, even for the models commonly used for survival data, i.e. Weibull or Cox regression, the log link function is used to connect the linear predictor to the hazard function of the target variable, and the effect scale is the same as before.

**3.3.4 Correlation.** Before displaying formulas for a correlation, let us discuss its suitability as an effect. The related question is: "Is there a (monotonic, or even linear) relationship between the variables $Y^{(1)}$ and $Y^{(2)}$?" According to the basic theme, we need to insert the word "relevant" into this question. But this does not necessarily make the question relevant. What would be the practical use of knowing that there is a relationship? It may be that

- there is a causal relationship; then, the problem is one of simple regression, as just discussed, since the relationship is then asymmetric, from a cause $X$ the a response $Y$;

- one of the variables should be used to infer ("predict") the values of the other; again a regression problem;

- in an exploratory phase, the causes of a relationship may be indirect, both variables being related to common causes, and this should lead to further investigations; this is then a justified use of the correlation as a parameter, which warrants its treatment here.

The Pearson correlation is

$$\rho \quad = \quad \frac{\sum_{12}}{\sqrt{\sum_{11}\sum_{22}}} \,, \quad \sum_{jk} = \varepsilon\left((Y^j - \varepsilon(Y^j))(Y^k - \varepsilon(Y^k))\right) \,.$$

A suitable effect scale is given by Fisher's well-known transformation

$$\vartheta = \frac{1}{2}\text{logit}((\rho + 1)/2) = \frac{1}{2}\log\left((1+\rho)/(1-\rho)\right) \,, \tag{6}$$

which extends the limited range of values of $\rho$ to all real numbers as it does in the case of proportions. When large correlations are compared, the effect as measured by the difference of $\vartheta$ values is approximately $\vartheta = \vartheta_1 - \vartheta_0 \approx \frac{1}{2}\log\left((1-\rho_0)/(1-\rho_1)\right)$, that is, it compares the complements to the correlation on a relative (logarithmic) scale. For correlations around zero, the effect turns out to be approximately equal to the correlation itself and to the effect for the regression coefficient.

# 4 General multivariate effects and multiple regression

This section is technically more involved. Readers are encouraged to continue with Section 5 in a first run.

## 4.1 The general model

The models just discussed are special cases of the general parametric model

$$\underline{Y}_i \sim F(\underline{\theta}, \underline{\phi}; \underline{x}_i) \,, \tag{7}$$

where $\underline{\theta}$ is the parameter of interest, $\underline{\phi}$ denotes nuisance parameters, and the distribution $\mathcal{F}$ may vary between observations depending on covariates $\underline{x}_i$. The parameters and covariates may be multidimensional. Interest is in a suitable function $\underline{\vartheta} = \underline{g}(\underline{\theta})$ that turns the parameter of interest into the effect as measured on the "effect scale" if desired. Of course, $\underline{\vartheta}$ may be $\underline{\theta}$ without transformation. There is typically a value $\underline{\vartheta}_0$, and the question is if the true $\underline{\vartheta}$ differs from it to a relevant extent. If $\vartheta$ is one-dimensional, the interest is in differences in one direction, $\vartheta > \vartheta_0$, say, and there is a threshold $\zeta$ defining the relevance.

**4.1.1 Standardization.**   A natural way to standardize the effect parameter generalizes the idea of Cohen's *d* to compare the effect to a kind of scatter in the observations. The contribution of a single observation to the inherent uncertainty of estimating the effect is given by the Fisher information

$$
\begin{aligned}
\mathbf{J}_i(\underline{\theta}, \underline{\phi}) &= \int s(\underline{y}; \underline{x}_i; \underline{\theta}, \underline{\phi}) s(\underline{y}; \underline{x}_i; \underline{\theta}, \underline{\phi})^{\mathsf{T}} \, dF(\underline{y}; \underline{x}_i; \underline{\theta}, \underline{\phi}) \\
s(\underline{y}; \underline{x}_i; \underline{\theta}, \underline{\phi}; \underline{x}_i) &= \partial \log \left( f(\underline{y}; \underline{x}_i; \underline{\theta}, \underline{\phi}) \right) / \partial(\underline{\theta}, \underline{\phi}) \,,
\end{aligned} \tag{8}
$$

where *f* is the density of the distribution *F*. The inverse of

$$\mathbf{J}(\underline{\theta}, \underline{\phi}) = \operatorname{ave}_i(\mathbf{J}_i(\underline{\theta}, \underline{\phi}))$$

equals *n* times the asymptotic variance-covariance matrix of the maximum likelihood estimator $\hat{\theta}_{ML}$. Therefore, if $\theta$ is one-dimensional,

$$\vartheta = \theta \sqrt{J(\theta, \underline{\phi})} \approx \theta / \sqrt{n \operatorname{var} \hat{\theta}_{ML}}$$

is the announced standardized effect.

**4.1.2 Effect norm.**   In the general case, $\underline{\theta}$ is multidimensional and the interest is in a function $\underline{\vartheta} = \underline{g}(\underline{\theta})$. In the regression case to be discussed below, $g$ will just select components of $\underline{\theta}$. A plausible general way to formalize the relevance for a *p*-dimensional $\underline{\vartheta}$ is based on a matrix $\mathbf{Q}$ that defines the norm $\eta$ by

$$\eta^2 = (\underline{\vartheta} - \underline{\vartheta}_0)^{\mathsf{T}} \mathbf{Q} \, (\underline{\vartheta} - \underline{\vartheta}_0)/p \tag{9}$$

and the question is if $\eta$ exceeds the relevance threshold $\zeta$. The natural choice of $Q$ is then

$$\mathbf{Q} = \mathbf{B} \, \mathbf{J}(\theta, \underline{\phi}) \mathbf{B}^{\mathsf{T}} \,, \quad \mathbf{B} = \partial \underline{\vartheta} / \partial \underline{\theta} \,.$$

**4.1.3 Inference.**   As mentioned in Section 3.1, these considerations only concern parameters and therefore, estimation methods are needed to get point and interval estimates in applications. Whereas such estimators $(\hat{\underline{\theta}}, \hat{\underline{\phi}})$ usually are approximately multivariate normal, $p\eta^2$ then follows approximately a chi-squared distribution or a mixture of scaled chi-squares.

**4.1.4 Multidimensional effect?.** Note that in this treatment of the problem, the alternative hypothesis is no longer one-sided for the parameter of interest itself—although it is, for $\eta$—, since there is no natural ordering in the multivariate space. This shows an intrinsic difficulty of the present approach for multivariate effects. However, the limitation mirrors the difficulty of asking scientifically relevant questions to begin with: What would be an effect that leads to new scientific insight?

In order to fix ideas, let us consider a regression model with a multivariate target variable. For example, $\underline{Y}$ may be a characterization of color or of shape, and the multivariate regression model may describe the effect of a treatment on the expected value of $\underline{Y}$. In the case of a single predictor, e.g., in a two-groups situation, the parameter of interest $\underline{\theta}$ has a direct interpretation as the difference of colors, shapes or the like, and a range of relevant differences may be determined using a norm that characterizes distinguishable colors or shapes, which will be different from **V**. In more general situations, it seems difficult to define the effect in a way that leads to a practical interpretation.

If the target variable $\underline{Y}$ measures different aspects of interest, like quality, robustness and price of a product or the abundance of different species in an environment, the scientific problem itself is a composite of problems that should be regarded in their own right and treated as univariate problems in turn.

A more common situation where there is an intrinsically multidimensional effect comes up in regression for a single target variable with categorical predictor variables in regression, to be discussed now.

## 4.2 Multiple regression and analysis of variance

In the multiple regression model, the predictor is multivariate,

$$Y_i = \alpha + \underline{x}^{i\top}\underline{\beta} + \varepsilon_i \, , \qquad \varepsilon_i \sim \mathcal{N}(0, \sigma^2) \, . \tag{10}$$

The model also applies to (fixed effects) analysis of variance or general linear models, where a categorical predictor variable (often called a factor) leads to a group of components ("dummy variables") in the predictor vector $\underline{x}_i$ and correspondingly in the coefficient vector $\underline{\beta}$.

Since we set out to ask scientifically relevant questions, a distinction must be made between two fundamentally different situations in which the model is proposed.

(a). In technical applications, the $\underline{x}$ values are chosen by the experimenter and are therefore fixed numbers. Then, a typical question is whether changing the values from an $\underline{x}_0$ to $\underline{x}_1$ evokes a relevant change in the target variable $Y$. This translates into the relevance of a single *coefficient* $\beta_j$ or of several of them.

(b). In other fields of applications, the values of the predictor variables are often also random, and there is a joint distribution of $\underline{X}$ and $Y$. A very common type of question asks whether a predictor variable or a group of them have a relevant influence on the target variable. The naive interpretation of influence here is that, as in the foregoing situation, an increase of the variable $X^{(j)}$ by one unit leads to a change given by $\beta_j$ in the target variable $Y$. However, this is not necessarily true since even if such an intervention may be possible, it can cause changes in the other predictors that lead to a compensation or an enhancement of the effect described by $\beta_j$. Thus, the question if $\beta_j$ is relevantly different from 0 is of unclear scientific merit.

A related question asks if a predictor (a component $x^{(j)}$) contributes in a relevant manner to the "explanatory value" of the model. This extends naturally to a group of coefficients that constitute a "term" of the model, typically a categorical predictor. In other words,

one asks if the effect of *dropping* the predictor or the term from the complete model is relevant.

Another legitimate use of the model is estimation of an unknown value of the response $Y$, $Y_0$, on the basis of known values $\underline{x}_0 0$ of the predictors, usually called *"prediction."* Then, one may ask if a predictor or a group of them reduce the prediction error by a relevant amount.

It is of course also legitimate to use the model as a description of a dataset. Then, statistical inference is not needed, and there is a high risk of over-interpretation of the outputs obtained from the fitting functions.

(c).  An intermediate situation can occur if the researcher can select observation units that differ mainly in the values of a given subset of predictor variables. Then, any remaining predictors should be excluded from the model, and the situation can be interpreted, with caution, as in the experimental situation.

**4.2.1 Coefficient effect.**   Let us first consider the experimental situation, where the effect of interest is a part of $\underline{\beta}$. If it reduces to a single coefficient $\beta_j$, the other components are part of $\underline{\phi}$, and the formulas for simple regression generalize in a straightforward way. The "coefficient effect" is

$$\vartheta_j = \delta_j \beta_j / \sigma \; ,$$

where $\delta_j$ is the empirical standard deviation $s_j$ for a continuous $x^{(j)}$ and half the difference between the two possible values if $x^{(j)}$ is binary.

**4.2.2 Drop effect.**   Applying the concept of standardization introduced above for the general model (7) leads to

$$\vartheta_{\mathrm{drop},j} = \beta_j / \sqrt{n \, \mathrm{var}(\hat{\beta}_{j,ML})} = \vartheta_j \kappa_n \sqrt{1 - R_j^2} \; , \tag{11}$$

where the second equation holds if there is an intercept in the model and its coefficient is not $\beta_j$, $R_j$ is the multiple correlation of the predictor $X^{(j)}$ with all the other predictors, and $\kappa_n = \sqrt{1 - 1/n} \approx 1$. The proof is given in the S1 Appendix.

Eq (11) shows that $\vartheta_{\mathrm{drop},j}$ turns into the test statistic of the t-test for dropping the predictor $X^{(j)}$ from the model, divided by $\sqrt{n}$, if estimators are plugged in for the parameters—whence its name "drop effect." It measures the change in the response (in $\sigma$ units) of increasing $X^{(j)}$, orthogonalized on the other predictors, by one of its standard deviations. If the predictor $X^{(j)}$ is orthogonal to the others, $\vartheta_{\mathrm{drop},j}$ and $\vartheta_j$ coincide.

If a categorical predictor is in the focus, a contrast between its levels may be identified as the effect of interest. For example, a certain group may be supposed to have higher values for the target variable than the average of the other groups. Then, the problem can be cast in the same way as for the single coefficient.

**4.2.3 Multidimensional drop effect.**   The effect of a categorical variable or another term in the model giving rise to a set $\underline{\beta}_J$ of cocefficients $\beta_j, j \in J$, can be assessed as a multidimensional effect. The general model (7) leads to

$$p \, \eta_J^2 = (\underline{\beta}_J - \underline{\vartheta}_0)^{\mathsf{T}} ((\mathbf{C}^{-1})_{JJ})^{-1} (\underline{\beta}_J - \underline{\vartheta}_0) / \sigma^2 \; , \quad \mathbf{C} = \mathrm{ave}_i \underline{x}_i \underline{x}_i^{\mathsf{T}} \; . \tag{12}$$

The derivation is again deferred to the S1 Appendix.

Noting that $\sigma^2(\mathbf{C}^{-1})_{JJ}/n$ is the variance-covariance matrix of the estimated effect $\hat{\underline{\beta}}_J$ makes again clear that $\eta_J$ is the norm of a kind of standardized effect, and that $n\,\eta_J^2$ is related to the F test statistic for examining if the term can be dropped from the model.

**4.2.4 Prediction effect.** The prediction error for predicting $Y_0$ for a given predictor vector $\underline{x}_0$ has two sources: the variability of the predicted value, which depends on the observations used for estimating the parameters, and the random deviation $\epsilon_0$ that is intrinsic to making the new observation $Y_0$. The latter is characterized by the parameter $\sigma$ and will be considered here. The question to be asked is: Is the reduction in the random variation $\sigma$ obtained by using a group of predictors relevant? The model with the group, called the "full model," is compared with the "reduced model," without them, and the corresponding $\sigma$'s are $\sigma_f$ and $\sigma_r$. The following technical comment defines the parameters precisely.

**4.2.5 Remark.** Whereas $\sigma_f$ is the $\sigma$ of the regression model (10), $\sigma_r$ needs a definition as will $\sigma_Y$ below. Assuming that (10) is correct, the reduced model will push some effects—that is, some constants—into the error term. The model results from projecting $\mathbf{X}\underline{\beta}$ (where $\mathbf{X}$ is the design matrix collecting the $\underline{x}_i$'s as its rows) to the space spanned by the reduced design matrix $\mathbf{X}_r$, with projection matrix

$$\mathbf{H}_r = \mathbf{X}_r^{\top}(\mathbf{X}_r^{\top}\mathbf{X}_r)^{-1}\mathbf{X}_r^{\top} \;.$$

Therefore, let $\gamma = (\mathbf{X}_r^{\top}\mathbf{X}_r)^{-1}\mathbf{X}_r\mathcal{E}(\underline{Y})$ be the linear fit to the expected values of $Y$. Then,

$$\mathbf{X}_r\underline{\gamma} = \mathcal{E}(\mathbf{H}_r\underline{Y}) = \mathbf{H}_r\mathbf{X}\underline{\beta} \quad \text{and} \quad \sigma_r^2 = \frac{1}{n}\mathcal{E}((\underline{Y} - \mathbf{X}_r\underline{\gamma})^{\top}(\underline{Y} - \mathbf{X}_r\underline{\gamma})) \;.$$

Below, we will need $\sigma_Y^2$, defined as

$$\sigma_Y^2 = \mathrm{ave}_i\,\mathcal{E}((Y_i - \mu)^2) \;, \quad \mu = \mathrm{ave}_i\,\underline{x}_i\underline{\beta} \;, \qquad ,$$

although this definition is a place holder since $\sigma_Y^2$ will cancel in the definition of the effect. Alternatively to these definitions, the model may be modified by assuming $\underline{x}_i$ to be random, with arbitrary distribution. Then, averages should be replced by expectations.

In the sequel, I will use the multiple correlation $R$, related to the variances of the random deviations and of $Y$ by

$$R^2 = 1 - \sigma^2/\sigma_Y^2 \;, \qquad \sigma^2 = (1 - R^2)\,\sigma_Y^2 \;,$$

where $\sigma_Y$ is the (marginal) standard deviation of $Y$ (see the remark for an exact definition), and $J$ collects the predictors that do not appear in in the reduced model.

A comparison of variances—or other scale parameters for that matter—is best done in the logarithmic scale, since relative differences are a natural way of expressing such differences (cf. Section 3.1). Then, an effect measure is

$$\vartheta_{\mathrm{pred},J} = \log\left(\sigma_r/\sigma_f\right) = \frac{1}{2}\log\left(\theta_J\right) \;, \qquad \theta_J = \frac{\sigma_r^2}{\sigma_f^2} = \frac{1 - R_r^2}{1 - R_f^2} \;. \tag{13}$$

It measures the log (-percent) increase in the error standard deviation caused by dropping the considered group of predictors from the model.

For simple analysis of variance, the model for comparing several groups, $\theta_J$ reduces to $\theta = 1/(1 - R_f^2)$, where $R_f^2$ is the fraction of the target variable's variance explained by the grouping, called $\eta^2$ in [16] and is between 0 and 1.

Note that $\vartheta_{\text{pred},J} = \tilde{g}(R_r) - \tilde{g}(R_f)$, where

$$\tilde{g}(R) = -\frac{1}{2} \, \log\left(1 - R^2\right) .$$

It is related to Fisher's z transformation $g$ for correlations (6) by $\tilde{g}(R) = g(R) - \log(1 + R)$ and shows the same behavior for large $R$.

In fact, the prediction effect is closely related to the drop effect, since

$$\theta_J = 1 + p \, \eta_J^2$$

as shown in the S1 Appendix. Thus,

$$\vartheta_{\text{pred},J} = \frac{1}{2} \, \log\left(1 + \vartheta_a^{*2}\right) \approx \frac{1}{2} \vartheta_a^{*2} , \tag{14}$$

the approximation being useful for reasonably small $\vartheta_a^*$.

**4.2.6 Estimation.** The effects $\eta_J$ and $\vartheta_{\text{pred},J}$ are estimated by plugging in estimates for the parameters, $\hat{\underline{\beta}}_J$, $\hat{\sigma} = \hat{\sigma}_f$ and $\hat{\sigma}_r$, into (12) or (13). Using the first option shows how to obtain a confidence interval. Assume that $\underline{\vartheta}_0 = \underline{0}$ as is almost always the case. (It can always be achieved by subtracting $\mathbf{X}_J \underline{\vartheta}_0$ from both sides of the model (10) and renaming $Y$ and $\underline{\beta}$.) Then, if the true $\underline{\beta}_J$ is $\underline{0}$, $n \hat{\eta}_J^2$ has a (central) $F$ distribution with $p$ and $v$ degrees of freedom, where $v$ is the number of degrees of freedom for $\hat{\sigma}$. If $\underline{\beta}_J$ is non-zero, this leads to a non-centrality of $np \eta_J^2$. Let $c_0$ and $c_1$ be the solutions of

$$F(X < n \hat{\eta}_J^2; p, v; c_0) = 0.975 , \quad F(X < n \hat{\eta}_J^2; p, v; c_1) = 0.025 ,$$

respectively, where $c_0 = 0$ if there is no solution for the first equation. Then $[c_0, c_1]/(np)$ is the confidence interval for $\eta_J$.

**4.2.7 Which effect measure?.** There are now three versions of effect measures for a term of a regression model:

- The coefficient effect $\vartheta_j = \beta_j \, \delta_j / \sigma$ describes the effect of manipulating a single predictor variable $X^j$.

- The drop effect examines the importance of a term involving a single coefficient, $\vartheta_{\text{drop},j}$, or several of them, $\eta_J$, for "explaining" the target variable $Y$. If the term is orthogonal to the other terms in the model, $\vartheta_{\text{drop},j}$ coincides with $\vartheta_j$.

- The prediction effect $\vartheta_{\text{pred},J}$ is a function of $\eta_J$ and measures the effect of a term on reducing the essential part of the prediction error, the standard deviation of the random error $\epsilon_i$, on the logarithmic scale.

Programs should provide the three measures for each term in a model. The scientific question should determine which one is appropriate for interpretation (see (a) and (b) above).

## 5 Relevance thresholds

The arguments in the Introduction have lead to the molesting requirement of choosing a threshold of relevance, $\zeta$. Ideally, such a choice is based on the specific scientific problem under study. However, researchers will likely hesitate to take such a decision and to argue for it. Conventions facilitate such a burden, and it is foreseeable that rules will be invented and adhered to sooner or later, analogously to the ubiquitous fixation of the testing level $\alpha = 5\%$.

Therefore, some considerations about simple choices of the relevance threshold in typical situations follow here.

### 5.1 One and two samples, regression coefficients

An established "small" value of Cohen's $d$ is 20% [13]. It may serve as the threshold for $d$. Since $d = 2\vartheta$ in the case of two groups (5), this leads to $\zeta = 10\%$ for $\vartheta$, which can be used also for a single sample and the coefficient effect in regression according to the discussion in Section 3. It extends to drop effects for terms with a single degree of freedom because they coincide if $R_j = 0$ (11), and from there to multivariate drop effects. However, this threshold transforms to a tiny effect $\vartheta_{\mathrm{pred},J}$ of $0.5\%\ell$ on the log ratio of lengths of prediction intervals according to (14). A threshold of $5\%\ell$ seems be more appropriate here. This shows again that the scientific question should guide the choice of the effect scale and of the relevance threshold!

### 5.2 Relative effect

General intuition may often lead to an agreeable threshold expressed as a percentage. For example, for a treatment to lower blood pressure, a reduction by 10% may appear relevant according to common sense. Admittedly, this value is as arbitrary as the 5% testing level. Physicians should determine if such a change usually entails a relevant effect on the patients' health, and subsequently, a corresponding standard might be generally accepted for treatments of high blood pressure.

As discussed in Section 3.1, when percentage changes are a natural way to describe an effect, it is appropriate to express it formally on the log scale, like $\vartheta = \varepsilon(\log(Y^{(1)})) - \varepsilon(\log(Y^{(0)}))$ in the two samples situation. Then, one might set $\zeta = 0.1$ for a 10% relevance threshold for the change, or more precisely, using the "log percent" scale, as $\zeta = 10\%\ell$.

### 5.3 Log-linear models

Several useful models connect the logarithm of the expected response with a linear combination of the predictors, notably Poisson regression with the logarithm as the canonical link function, log-linear models for frequencies, and Weibull regression, a standard model for reliability and survival data. Here, the consideration of a relative effect applies again. An increase of 0.1 in the linear predictor leads to an increase of 10% in the expected value, and therefore, $\zeta = 10\%\ell$ seems appropriate for the standardized coefficients $\vartheta_j = \beta_j\, s_j$.

### 5.4 Proportions and logistic regression

As the "logit percent" scale (Section 3.2) extends the log percent scale and matches it for small proportions, the same threshold $\zeta = 10\%\ell$ should be applied. It declares a difference between $p = 0.5$ and $p = 0.525$, or between 0.35 and 0.373, or between 0.1 and 0.109 as relevant. Like for the log-linear models, this threshold also applies to standardized coefficients in logistic regression.

### 5.5 Correlation

In the two samples situation, considering the $x_i$ as random and assuming equal probabilities for both groups, the correlation is

$$\rho^2 = (d/2)^2 / (1 + (d/2)^2) ,$$

and the threshold of 20% on Cohen's $d$ leads again to $\zeta = 0.1$. In Section 3, the logit scale according to Fisher has been recommended as the effect scale. Since $\vartheta = g(\rho) \approx \rho$ for $\rho \leq 0.1$,

**Table 2. Models, recommended effect scales and relevance thresholds.**

| Problem | Basic model | Effect $\vartheta = g(\theta)$ | Rel. thresh. $\zeta$ |
|---|---|---|---|
| One, or two paired samples | $\mathcal{N}(\mu, \sigma^2)$ | $\mu/\sigma$ | 10% |
| Two independent samples | $\mathcal{N}(\mu_k, \sigma^2)$ | $d = (\mu_1 - \mu_0)/\sigma$ <br> $\vartheta = d/2$ | 20% <br> 10% |
| Regression coefficient effect <br><br> drop effect <br> prediction effect | $Y_i = \alpha + \underline{x}^{i^T}\underline{\beta} + \epsilon_i$ <br> $\epsilon_i \sim \mathcal{N}(0, \sigma^2)$ | $\beta_j \delta_j / \sigma$ <br> $\eta_J$ <br> $-\frac{1}{2}\log(1 - R^2)$ | 10% <br> 10% <br> 0.5%$\ell$ or 5%$\ell$ |
| Relative Difference | $\log(Y) \sim \mathcal{N}(\mu_k, \sigma^2)$ | $\log(\mu_1/\mu_0)$ | 10%$\ell$ |
| Proportion | $\mathcal{B}(n, p)$ | $\log(p/(1-p))$ | 10%$\ell$ |
| Logistic regression | $\mathrm{logit}(P(Y_i = 1)) = \alpha + \underline{x}^{i^T}\underline{\beta}$ | $\underline{\beta}_j s_j$ | 10%$\ell$ |
| Correlation | $\underline{Y} \sim \mathcal{N}_2(\underline{\mu}, \sum)$ <br> $\rho = \sum_{12}/\sqrt{\sum_{11}\sum_{22}}$ | $\frac{1}{2}\log\left(\frac{1+\rho}{1-\rho}\right)$ | 10%$\ell$ |

the threshold can be used on this scale, too, and the "logit percent" notation is appropriate, $\zeta = 10\%\ell$.

## 5.6 Summary

The scales and thresholds for the different models that are recommended here for the case that the scientific context does not suggest any choices are listed in Table 2.

# 6 Description of results

It is common practice to report the statistical significance of results by a p-value in parenthesis, like "The treatment has a significant effect ($p = 0.04$)," and estimated values are often decorated with asterisks to indicate their p-values in symbolized form. If such short descriptions are desired, secured relevance values should be given. If Rls $> 1$, the effect is relevant, if it is $> 0$, it is significant in the traditional sense, and these cases can be distinguished in even shorter form in tables by plusses or an asterisk as symbols as follows: * for significant, that is, Rls $> 0$; + for relevant (Rls $> 1$); ++ for Rls $> 2$; and +++ for Rls $> 5$. To make these indications well-defined, the relevance threshold $\zeta$ must be declared either for a whole paper or alongside the indications, like "Rls $= 1.34$ ($\zeta = 10\%\ell$)." Since the secured (and potential) relevance also depends on the confidence level $1 - \alpha$, this quantity should also be declared.

## 6.1 Examples

The first examples are taken from the first "manylabs" project about replicability of findings in psychology [17], since for that study, the scientific questions had been judged to deserve replication and full data for the replication is easily available.

The original studies were replicated in each of 36 institutions. Here, I pick the replication at Penn State University of the following item: "Students were asked to guesstimate the height of Mount Everest. One group was 'anchored' by telling them that it was more than 2000 feet, the other group was told that it was less than 45,500 feet. The hypothesis was that respondents would be influenced by their 'anchor,' such that the first group would produce smaller numbers than the second" [18]. The true height is 29,029 feet.

**Table 3. Data for the anchoring example in 1,000 feet.**

| group "low", $n_0 = 8$ | 2.3 | 2.7 | 3 | 3 | 3.1 | 6 | 12 | 15 | | | | |
|---|---|---|---|---|---|---|---|---|---|---|---|---|
| group "high", $n_1 = 12$ | 25 | 32 | 34 | 40 | 40 | 40 | 42.7 | 43.5 | 44 | 45 | 45 | 45.5 |

According to the discussion in Section 5, the data is analyzed here on the logarithmic scale, and the threshold of $10\%\ell$ is applied. The data, reduced to the first 20 observations for simplicity, are given in Table 3.

The group means of the log values are 1.52 and 3.67 (corresponding to 4,560 and 39,190 feet) and the standard error for their difference is 0.216 on 18 degrees of freedom. This leads to a confidence interval of $\hat{\vartheta} \pm \hat{\omega} = 2.15 \pm (2.10 \cdot 0.216) = [1.70, 2.60]$ and $\text{Sig0} = \hat{\vartheta}/\hat{\omega} = 4.74$. The relevance is $\text{Rle} = 100 \cdot \hat{\vartheta}/\zeta = 2.15/0.1 = 21.5$ with interval limits of $\text{Rls} = \text{Rle} - \hat{\omega}/\zeta = 17.0$ and $\text{Rlp} = \text{Rle} + \hat{\omega}/\zeta = 26.0$. The single value notation is $\text{Rls} = 17.0$ ($\zeta = 10\%\ell$). This is an extremely clear effect.

A second study asked if a positive or negative formulation of the same options had an effect on the choice [19]. Confronted with a new contagious disease, the government has a choice between action A that would save 200 out of 600 people or action B which would save all 600 with probability 1/3. The negative description was that either (A) 400 would die or (B) all 600 would die with probability 2/3. I report the results for Penn State (US) and Tilburg (NL) universities. The data is summarized in Table 4, and the effect, significance, and relevance, in Table 5. The secured relevance is $\text{Rls} = 4.16$ ($\zeta = 10\%\ell$) and 10.1 ($\zeta = 10\%\ell$) for the two institutions, the effect is thus clearly relevant. One may ask if there is a relevant (!) difference between these two studies, with a view of applying the notions of this paper to the theme of replicability. This will be done in a forthcoming paper.

The third example is a multiple regression problem. The dataset reflects the blasting activity needed for digging a freeway tunnel beneath a Swiss city. Since blasting can cause damage in houses located at a small `distance` from the point of blasting, the `charge` should be adjusted to keep the `tremor` in the basement of such a house below a threshold $y_0$. The logarithmic `tremor` is modelled as a linear function of the logarithmic `distance` and `charge`, an additive adjustment to the house where the measurements are taken (factor `location`), and `time`, a rescaled calendar day. Only part of the data for 3 locations are used here, see Table 6.

**Table 4. Data for the second example.**

| | PSU | | | Tilburg | | |
|---|---|---|---|---|---|---|
| | *n* | A | B | *n* | A | B |
| negative | 48 | 16 | 32 | 34 | 6 | 28 |
| positive | 47 | 30 | 17 | 46 | 29 | 17 |

**Table 5. Results for the second example.** Relevance threshold $10\%\ell$.

| | effect | | | signif. | relevance | | |
|---|---|---|---|---|---|---|---|
| | $\hat{\theta}$ | low | high | Sig0 | Rle | Rls | Rlp |
| PSU | **1.26** | 0.42 | 2.10 | 1.49 | 12.6 | **4.2** | 21.0 |
| Tilburg | **2.08** | 1.01 | 3.14 | 1.95 | 20.8 | **10.1** | 31.4 |

**Table 6. Data for the blasting example.**

| charge | dist | loc'n | time | tremor | charge | dist | loc'n | time | tremor |
|---|---|---|---|---|---|---|---|---|---|
| 4.760 | 62 | loc1 | 0.5562 | 4.07 | 3.640 | 55 | loc1 | 0.7644 | 4.31 |
| 4.848 | 58 | loc1 | 0.5699 | 0.71 | 3.708 | 61 | loc1 | 0.7699 | 4.43 |
| 5.824 | 55 | loc1 | 0.5890 | 6.71 | 3.812 | 46 | loc2 | 0.7726 | 10.67 |
| 6.656 | 50 | loc1 | 0.6082 | 12.23 | 3.725 | 69 | loc4 | 0.7808 | 2.00 |
| 6.656 | 42 | loc1 | 0.6274 | 10.55 | 3.305 | 67 | loc1 | 0.7836 | 2.51 |
| 4.368 | 37 | loc1 | 0.6384 | 16.90 | 3.744 | 50 | loc2 | 0.7863 | 7.91 |
| 5.200 | 33 | loc1 | 0.6548 | 16.90 | 3.725 | 65 | loc4 | 0.7863 | 3.47 |
| 4.998 | 31 | loc1 | 0.6685 | 14.99 | 3.725 | 55 | loc2 | 0.7918 | 5.63 |
| 4.998 | 49 | loc2 | 0.6712 | 8.39 | 3.870 | 61 | loc4 | 0.8000 | 2.36 |
| 5.236 | 29 | loc1 | 0.6849 | 16.42 | 4.765 | 60 | loc2 | 0.8055 | 6.59 |
| 5.593 | 44 | loc2 | 0.6877 | 12.23 | 1.248 | 59 | loc4 | 0.8055 | 1.70 |
| 1.190 | 30 | loc1 | 0.6904 | 5.03 | 4.644 | 62 | loc2 | 0.8082 | 5.15 |
| 4.998 | 41 | loc2 | 0.6932 | 12.23 | 5.285 | 56 | loc4 | 0.8110 | 5.39 |
| 4.998 | 31 | loc1 | 0.7041 | 14.27 | 5.285 | 69 | loc2 | 0.8219 | 5.27 |
| 5.712 | 38 | loc2 | 0.7068 | 23.38 | 0.624 | 53 | loc4 | 0.8247 | 1.07 |
| 4.680 | 35 | loc1 | 0.7123 | 13.91 | 3.986 | 73 | loc2 | 0.8274 | 5.03 |
| 4.702 | 36 | loc2 | 0.7233 | 14.15 | 2.490 | 51 | loc4 | 0.8301 | 4.43 |
| 4.784 | 39 | loc1 | 0.7260 | 9.95 | 4.390 | 79 | loc2 | 0.8411 | 4.43 |
| 5.824 | 36 | loc2 | 0.7288 | 13.43 | 4.390 | 50 | loc4 | 0.8438 | 5.99 |
| 4.160 | 43 | loc1 | 0.7425 | 10.55 | 3.870 | 85 | loc2 | 0.8466 | 2.63 |
| 3.952 | 36 | loc2 | 0.7425 | 20.98 | 3.870 | 50 | loc4 | 0.8493 | 5.27 |
| 3.744 | 88 | loc4 | 0.7452 | 1.52 | 1.768 | 50 | loc4 | 0.8685 | 1.58 |
| 3.194 | 50 | loc1 | 0.7479 | 7.07 | 2.496 | 51 | loc4 | 0.0000 | 3.29 |
| 3.744 | 38 | loc2 | 0.7507 | 14.51 | 3.640 | 52 | loc4 | 0.0192 | 4.67 |
| 3.305 | 79 | loc4 | 0.7616 | 1.43 | | | | | |

Results are collected in Table 7. The `time` does not show any significance and therefore no relevance either. The relevances of the coefficient and drop effects are related by (11). Thus, their ratio equals $\sqrt{1 - R_j^2}$ and is a useful measure of collinearity.

For the shortest description, the coefficient of `log10(charge)` would be indicated as $0.752^{+++}$.

The results for these examples have been otained by the R package `relevance`, available from "CRAN" https://r-project.org.

**Table 7. Extensive results for the blasting expample `location` is a factor with 3 levels.** Relevance thresholds are 10% for standardized coefficients and 5%ℓ for the prediction effect. The columns shown in bold face should be routinely shown.

| term | coef. | df | se | Sig0 | p.value | coef. | stand. Rlp | coef. effect Rls | drop effect Rlp | Rls | prediction eff. Rlp | Rls |
|---|---|---|---|---|---|---|---|---|---|---|---|---|
| location | | 2 | | 1.56 | 1.27e-03** | | | | 5.02 | **1.36**$^+$ | 5.58 | 0.12. |
| log10(distance) | **-2.022** | 1 | 0.198 | -5.06 | 4.68e-13*** | -1.666 | **19.87** | **13.5**$^{+++}$ | 17.82 | **11.94**$^{+++}$ | 14.75 | 9.16$^{+++}$ |
| log10(charge) | **0.752** | 1 | 0.130 | 2.86 | 7.94e-07*** | 0.959 | **12.85** | **6.3**$^{+++}$ | 11.74 | **5.01**$^{+++}$ | 8.96 | 2.20$^{++}$ |
| time | **0.062** | 1 | 0.138 | 0.22 | 0.66 | 0.069 | **3.68** | -2.3$^-$ | 3.45 | **0.00** | 1.00 | 0.000 |

## 7 Relevance instead of p-values

The deficiencies of the common use of p-values has lead to a fierce debate and a flood of papers, often resulting in the vague conclusion that the accused concept should be used with caution.

Here, I have argued that the origin of the crisis roots deeper: The misuse of the p-value reflects a way to avoid the effort of asking relevant scientific questions to begin with. Typical problems in empirical research often concern a quantity like the effect of a treatment on a specified target variable. These problems are only well-posed if there is a threshold of relevance. I am not the first to advocate this requirement, I emphasize its importance again and develop it further into the novel measure of relevance. It is essential to keep in mind that the threshold should be determined only by the scientific problem and therefore should depend as little as possible on the design of the study that estimates the effect—it must not depend on the number of observations.

Some earlier proposals are also based on an idea of a threshold that widens the null hypothesis, as the "Second Generation P-Value" by Blume *et al.* [1], which was discussed above. Kruschke [7] uses similar ideas as this paper with a Bayesian approach (see footnote 2.3), and a referee suggested to draw conclusions on the base of the posterior probability of the effect exceeding the threshold, $\vartheta > \zeta$. However, none of these proposals, neither frequentist nor Bayesian, has yet been widely applied.

The paradigm of null hypothesis significance testing that is so well established asks for the choice of a threshold: the significance level $\alpha$ of the test, or the confidence level $1-\alpha$. In principle, $\alpha$ could be arbitrarily chosen, but tradition has fixed it at 5% for most scientific fields. The relevance threshold introduces yet another choice to be made. A careful selection should be sought in each scientific study. Since this is a cumbersome requirement, conventions have been proposed in this paper for the most common situations.

The traditional method to convey the assessment of an effect in a more informative way than the p-value is the confidence interval. Its downside is that it consists of two numbers that carry the measurement unit of the effect and are therefore not directly comparable between studies. The significance measure introduced here is a single, standardized number that conveys the essentials of the confidence interval. It depends, however, again on a given value of the effect. When this value is 0, the basic flaw of the p-value is inherited. Combining it with the relevance threshold is a necessary step to give an appropriate characterization of the relevance of a result.

The combination is best achieved by focussing on the confidence interval for the relevance measure, with boundaries called "secured" and "potential" relevance. The secured relevance Rls may even be used as a single number characterizing the knowledge gained about the effect of interest.

A conclusion from the p-value debate is that a simple yes-no decision about the result is misleading. Since our thinking likes categorization, I have introduced labels characterizing the comparison of the confidence interval with both the zero effect and the relevance threshold. It is defined on the basis of the two significance values $Sig_0$ and $Sig_\zeta$ or of the two relevance limits Rls and Rlp.

The significance and relevance measures and the classification are straightforward enhancements of concepts that are well established and ubiquously known. There is hope that they can form a new standard of presenting statistical results.

### 7.1 Replicability

The p-value debate is closely related to and often confounded with the reproducibility crisis. In fact, there is ample evidence that in several fields of science, when a statistical study is

replicated, a significant effect found in the original study turns out to be non-significant in the replication, thereby formally failing the fundamental requirement of reproducibility of empirical science. While many causes are suggested and found for such failures, prominent ones are tied to the problems with statistical testing and the p-value discussed in the Introduction. Here is the argument:

The p-value was originally advocated as a filter against publication of results that may be due to pure randomness. It was soon converted into a tool to generate "significant" results regardless of their scientific relevance. This leads to so-called selection bias: When many studies examine small true effects with limited precision, some of them will turn out significant by chance, will thus pass the filter and be published, whereas the non-significant ones will go unnoticed. These studies will have a low probability of being successfully replicated.

Clearly, using the criterion of a secured *relevant* effect (case Rlv) as a filter would reduce the frequency of phony results drastically, since the barely significant results would rarely pass it. A relevant result in this sense will usually have a high probability of showing at least a significant estimate (case Amb.Sig) and an estimated relevance above 1 upon replication—unless the precision is low or data snooping has been extensively applied to get it. A securely relevant result can be expected if the replication has sufficient power.

The concepts introduced here can be profitably applied to assess replications of results also in more depth, as will be shown in a forthcoming paper.

## 8 Conclusion

The p-value has been (mis-) used to express the results of statistical data analyses for too long, in spite of the extensive discussions about the bad consequences of this practice for science.

It is time to introduce a new concept for the presentation of the statistical inference for an effect under study. The measure of relevance introduced here is suitable to achieve this goal. It needs the choice of a relevance threshold for the effect of interest, a requirement posed by the desire to ask a scientifically meaningful question to begin with.

The goal of a typical statistical enquiry is to prove that an effect is relevant. Based on the measures "secured relevance," Rls, and "potential relevance," Rlp, either this can be achieved, or a "negligible" effect can be found—or the answer may be "ambiguous."

Application of these concepts will enhance reproducibility: When relevant effects are examined rather than merely significant ones, the replication will much more often turn out to be at least significant in the replication.

## Supporting information

**S1 Appendix.**
(PDF)

## Acknowledgments

Samuel Pawel of the Biostatistics group at University of Zurich and my colleagues Martin Mächler and Matteo Tanadini have provided valuable comments and suggestions on the writeup of this paper.

## Author Contributions

**Conceptualization:** Werner A. Stahel.

**Formal analysis:** Werner A. Stahel.

**Investigation:** Werner A. Stahel.

**Methodology:** Werner A. Stahel.

**Software:** Werner A. Stahel.

**Visualization:** Werner A. Stahel.

**Writing – original draft:** Werner A. Stahel.

**Writing – review & editing:** Werner A. Stahel.

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
