## [Decision Letter · Decision Letter 0]

1 Dec 2020

PONE-D-20-29052

Measuring Significance and Relevance Instead of P-values

PLOS ONE

Dear Dr. Stahel,

Thank you for submitting your manuscript to PLOS ONE. After careful consideration, we feel that it has merit but does not fully meet PLOS ONE’s publication criteria as it currently stands. Therefore, we invite you to submit a revised version of the manuscript that addresses the points raised during the review process.

The entire premise of your manuscript seems to be that the current system is not working.  That is, at best, a debatable point.  Back in 1998 I taught a class at Johns Hopkins with Jeff Blume, and we discussed these issues all the time.  I do have tremendous respect for him as an innovative thinker, but his arguments never did convince me that there is anything wrong with the p-value as it currently stands.  So that case still needs to be made, and I did not see a very strong case made here.  Yes, I do realize that some think of a p-value as the probability that the null hypothesis is true.  That is an indictment of their education, and not of the p-value itself.  Yes, we require more information than just the p-value alone.  And such additional information is pretty much always provided; never is a p-value produced in complete isolation.  Moreover, the additional information does not reproduce or render redundant the information contained in the p-value itself.  So, in the spirit of "If it ain't broke, don't fix it", I ask again ... what is wrong with the p-value?

We look forward to receiving your revised manuscript.

Kind regards,

Vance Berger

Academic Editor

PLOS ONE

Journal Requirements:

Reviewers' comments:

Reviewer's Responses to Questions

**Comments to the Author**

1. Is the manuscript technically sound, and do the data support the conclusions?

Reviewer #1: Yes

2. Has the statistical analysis been performed appropriately and rigorously? 

Reviewer #1: Yes

3. Have the authors made all data underlying the findings in their manuscript fully available?

Reviewer #1: Yes

4. Is the manuscript presented in an intelligible fashion and written in standard English?

Reviewer #1: Yes

5. Review Comments to the Author

Reviewer #1: This paper proposes a new method for evaluating an effect based on relevance rather on whether the effect is different from zero. The approach is appealing in its simplicity and Figure 2 shows how an effect might be graded in 1 of 6 categories rather than the usual significant vs not significant. The author makes a good argument that the proposed approach is better than TOST or SGPV.

Section 4 offers guidance on selecting a threshold of relevance. The author proposes to quantify the threshold using a formulaic approach. Any such quantity is arbitrary and won’t be applicable across all situations. This is illustrated in lines 495-499 where the author points to a situation in which the proposed threshold does not work and notes that “the scientific question should guide the choice of the effect scale and of the relevance threshold!” The author returns to this point in lines 596-598, and explains that he offered the formulaic approach because “conventions facilitate such a burden, it is foreseeable that they will be invented and adhered to sooner or later…therefore, considerations for such choices have been introduced in this paper.” It would be helpful if the author would provide that rationale at the beginning of Section 4, rather than leaving it to Section 6.

Section 5 offers guidance on reporting results. Here the simplicity of the approach is often lost, as the guidance suggests reporting values for a good number of items. Why not report the estimated parameter, its RLe, and its CI? The author makes the same argument in lines 608-609, but that is not what he has presented, for example, in Table 7.

Table 1. s and r are used in every row in the table but are undefined. If s a placeholder for Sig and r a placeholder for RL, it would have been clearer to use the appropriate column heading in place of s or r in the body of the table.

Line 226. I believe there is an error in using theta.

6. PLOS authors have the option to publish the peer review history of their article (what does this mean?). If published, this will include your full peer review and any attached files.

Reviewer #1: No

---

## [Author Response · Author response to Decision Letter 0]

7 Jan 2021

The responses to the editor and the reviewer are contained in 2 files

---

## [Decision Letter · Decision Letter 1]

1 Apr 2021

PONE-D-20-29052R1

New relevance and significance measures to replace p-values

PLOS ONE

Dear Dr. Stahel,

Thank you for submitting your manuscript to PLOS ONE. After careful consideration, we feel that it has merit but does not fully meet PLOS ONE’s publication criteria as it currently stands. Therefore, we invite you to submit a revised version of the manuscript that addresses the points raised during the review process.

If applicable, we recommend that you deposit your laboratory protocols in protocols.io to enhance the reproducibility of your results. Protocols.io assigns your protocol its own identifier (DOI) so that it can be cited independently in the future. For instructions see: http://journals.plos.org/plosone/s/submission-guidelines#loc-laboratory-protocols . Additionally, PLOS ONE offers an option for publishing peer-reviewed Lab Protocol articles, which describe protocols hosted on protocols.io. Read more information on sharing protocols at https://plos.org/protocols?utm_medium=editorial-email&utm_source=authorletters&utm_campaign=protocols .

We look forward to receiving your revised manuscript.

Kind regards,

Vance Berger

Academic Editor

PLOS ONE

Journal Requirements:

Reviewers' comments:

Reviewer's Responses to Questions

**Comments to the Author**

1. If the authors have adequately addressed your comments raised in a previous round of review and you feel that this manuscript is now acceptable for publication, you may indicate that here to bypass the “Comments to the Author” section, enter your conflict of interest statement in the “Confidential to Editor” section, and submit your "Accept" recommendation.

Reviewer #1: All comments have been addressed

Reviewer #2: All comments have been addressed

2. Is the manuscript technically sound, and do the data support the conclusions?

Reviewer #1: Yes

Reviewer #2: Partly

3. Has the statistical analysis been performed appropriately and rigorously? 

Reviewer #1: Yes

Reviewer #2: N/A

4. Have the authors made all data underlying the findings in their manuscript fully available?

Reviewer #1: Yes

Reviewer #2: Yes

5. Is the manuscript presented in an intelligible fashion and written in standard English?

Reviewer #1: Yes

Reviewer #2: Yes

6. Review Comments to the Author

Reviewer #1: (No Response)

Reviewer #2: See my attached file

7. PLOS authors have the option to publish the peer review history of their article (what does this mean?). If published, this will include your full peer review and any attached files.

Reviewer #1: No

Reviewer #2: No

---

## [Author Response · Author response to Decision Letter 1]

15 May 2021

I have attached the response to the referee's comments in the pdf file "response to reviewers"

---

## [Editor Report · Decision Letter 2]

27 May 2021

New relevance and significance measures to replace p-values

PONE-D-20-29052R2

Dear Dr. Stahel,

We’re pleased to inform you that your manuscript has been judged scientifically suitable for publication and will be formally accepted for publication once it meets all outstanding technical requirements.

Kind regards,

Vance Berger

Academic Editor

PLOS ONE
---

## [Editor Report · Acceptance letter]

4 Jun 2021

PONE-D-20-29052R2

New relevance and significance measures to replace p-values

Dear Dr. Stahel:

I'm pleased to inform you that your manuscript has been deemed suitable for publication in PLOS ONE. Congratulations! Your manuscript is now with our production department.

Kind regards,

on behalf of

Dr. Vance Berger

Academic Editor

PLOS ONE